# COMMUNICATING VIA MARKOV DECISION PROCESSES

## ABSTRACT

We consider the problem of communicating exogenous information by means of Markov decision process trajectories. This setting, which we call a Markov coding game (MCG), generalizes both source coding and a large class of referential games. MCGs also isolate a problem that is important in decentralized control settings in which cheap-talk is not available—namely, they require balancing communication with the associated cost of communicating. We contribute a theoretically grounded approach to MCGs based on maximum entropy reinforcement learning and minimum entropy coupling that we call greedy minimum entropy coupling (GME). We show both that GME is able to outperform a relevant baseline on small MCGs and that GME is able to scale efficiently to extremely large MCGs. To the latter point, we demonstrate that GME is able to losslessly communicate binary images via trajectories of Cartpole and Pong, while simultaneously achieving the maximal or near maximal expected returns, and that it is even capable of performing well in the presence of actuator noise.

## 1 INTRODUCTION

This work introduces a novel problem setting called Markov coding games (MCGs). MCGs are two-player decentralized Markov decision processes (Oliehoek et al., 2016) that proceed in four steps. In the first step, one agent (called the sender) receives a special private observation (called the message), which it is tasked with communicating. In the second step, the sender plays out an episode of a Markov decision process (MDP). In the third, the other agent (called the receiver) receives the sender's MDP trajectory as its observation. In the fourth, the receiver estimates the message from the received trajectory. The shared payoff to the sender and receiver is a weighted sum of the cumulative reward yielded by the MDP and an indicator specifying whether or not the receiver correctly decoded the message.

Among the reasons that MCGs are of interest is the fact that they generalize other important settings. The first of these is referential games. In a referential game, a sender attempts to communicate a message to a receiver using cheap talk actions—i.e., communicatory actions that do not have externalities on the transition or reward functions. Referential games have been a subject of academic interest dating back at least as far as Lewis's seminal work *Convention* (Lewis, 1969). Since then, various flavors of referential games have been studied in game theory (Skyrms, 2010), artificial life (Steels, 2013), evolutionary linguistics (Smith, 2002), cognitive science (Spike et al., 2017), and machine learning (Lazaridou et al., 2018). MCGs can be viewed as a generalization of referential games to a setting where we drop the often unrealistic assumption that the sender's actions do not incur costs.

A second problem setting generalized by MCGs is source coding (MacKay, 2002). In source coding (also known as data compression) the objective is to construct an injective mapping from a space of messages to the set of sequences of symbols (for some finite set of symbols) such that the expected output length is minimized. Source coding has a myriad of real world applications involving the compression of images, video, audio, and genetic data. MCGs can be viewed as a generalization of the source coding problem to a setting where the cost of an encoding may involve complex considerations, rather than simply being equal to the sequence length.

Yet another reason to be interested in MCGs is that they isolate an important subproblem of decentralized control. In particular, achieving good performance in an MCG requires the sender's actions

to simultaneously perform control in an MDP and communicate information (i.e., to communicate implicitly). This presents a challenge due to the fact that approximate dynamic programming, the foundation for preeminent approaches to constructing control policies (Sutton & Barto, 2018), is ill suited to constructing communication protocols because their values depend on counterfactuals. In other words, the information conveyed by an action depends on the policy at other contemporaneous states, violating the locality assumption of approximate dynamic programming approaches.

To address MCGs, we propose a theoretically grounded algorithm called greedy minimum entropy coupling (GME). GME leverages a union of maximum entropy reinforcement learning (MaxEnt RL) (Ziebart et al., 2008) and minimum entropy coupling (MEC) (Kovačević et al., 2015). The key insight is that maximizing the returns of the MDP can be disentangled from learning a good communication protocol by realizing that the entropy of a policy corresponds (in an informal sense) to its capacity to communicate. GME leverages this insight in two steps. In the first step, GME constructs a MaxEnt policy for the MDP, balancing between maximizing expected return and maximizing cumulative conditional entropy. In the second step, which occurs at each decision point, GME uses MEC to pair messages with actions in such a way that the sender selects actions with the same probabilities as the MaxEnt RL policy (thereby guaranteeing the same expected return from the MDP) and the receiver's uncertainty about the message is greedily reduced as much as possible.

To demonstrate the efficacy of GME, we present experiments for MCGs based on a gridworld, Cartpole, and Pong (Bellemare et al., 2013), which we call CodeGrid, CodeCart, and CodePong, respectively. For CodeGrid, we show that with a message space in the 10s or 100s, GME significantly is able to outperform a relevant baseline. For CodeCart and CodePong, we use a message space of binary images and a uniform distribution over messages, meaning that a randomly guessing receiver has an astronomically small probability of guessing correctly. Remarkably, we show that GME is able to achieve an optimal expected return in Cartpole and Pong while simultaneously losslessly communicating images to the receiver, demonstrating that GME has the capacity to be scaled to extremely large message spaces and complex control tasks. Moreover, we find that the performance of GME decays gracefully as the amount of actuator noise in the environment increases.

## 2   RELATED WORK

The works that are most closely related to this one can be taxonomized as coming from literature on referential games, source coding, multi-agent reinforcement learning, and diverse skill learning.

**Referential Games** Among work on referential games, Foerster et al. (2016)'s work is perhaps most similar in that it is concerned with directly optimizing the performance of a communication protocol. They propose DIAL, an algorithm that optimizes the sender's protocol by performing gradient ascent through the parameters of the receiver. Foerster et al. show that DIAL outperforms methods based on independent Q-learning on a variety of communication tasks. However, DIAL-based approaches are not directly applicable to MCGs, as they would require differentiating through trajectories.

**Coding** Another body of related work concerns extensions of the source coding problem. Length limited coding (Larmore & Hirschberg, 1990) considers a problem setting in which the objective is to minimize the expected sequence length (as before), subject to a maximum length constraint. Coding with unequal symbol costs (Golin et al., 2002; Iwata et al., 1997) considers the problem in which the goal is to minimize the expected cumulative symbol cost of the sequence to which the message is mapped. The cost of a symbol may differ from the cost of other symbols arbitrarily, making it a strictly more general problem setting than standard source coding (which can also be thought of as minimizing cumulative symbol cost with equally costly symbols). Both length limited coding and coding with unequal costs are subsumed by Markov coding games. And while existing algorithms for both standard source coding and the extensions above are well-established and widely commercialized, they are unable to address the more general MCG setting.

MCGs are also related to finite state Markov channel settings (Wang & Moayeri, 1995). In such settings, the fidelity of the channel by which the sender communicates to a receiver is controlled by a Markov process, which, in contrast to our work, transitions *independently* of the sender's decisions. Another related setting is intersymbol interference, where the sender's previously selected symbols (i.e., actions) may cause interference with subsequently selected symbols, making them less likely to be faithfully transmitted to the receiver (Lathi, 1998). MCGs differ from both Markov channel

and intersymbol interference settings in that the Markov system controls the cost paid by the sender, rather than interfering with the quality of the channel. MCGs are more resemblant of a setting in which the channel is reliable, but subject to natural variation in costs, such as based on weather or third party usage, as well as variation based on the sender's own usage.

**Multi-Agent Reinforcement Learning** A third related area comes from MARL literature. Strouse et al. (2018) investigate directly embedding a reward for taking actions with high mutual information into policy gradient objectives. They find that this approach can improve expected return in cooperative settings with asymmetric information. The baseline for our CodeGrid experiments loosely resembles Strouse et al.'s algorithm. More recently, Bhatt & Buro (2021) investigate an alternative approach whereby the sender's behavioral policy deterministically selects the action that maximizes the receiver's posterior probability of the correct message, when computed using the target policy. They show that this modification empirically yields significantly improved convergence properties as compared to other variations of independent reinforcement learning. However, this approach is not directly applicable to settings in which a single action must be used for both communication and control.

**Diverse Skill Learning** A fourth area of related research is that of diverse skill learning (Eysenbach et al., 2019). Eysenbach et al. (2019) propose an unsupervised learning method for discovering diverse, identifiable skills. Their objective, called DIAYN, seeks to learn diverse, discriminable skills. This paradigm resembles our work in the sense that skills can be interpreted as messages and discriminability can be interpreted as maximizing the mutual information between the skill and the state. The baseline used in our CodeGrid experiments can also be viewed as an adaptation of an idealized version of DIAYN to the MCG setting.

## 3 BACKGROUND AND NOTATION

We will require the following background and notation material to introduce Markov coding games and greedy minimum entropy coupling.

**Markov Decision Processes** To represent our task formalism, we use finite Markov decision processes (MDPs). We notate MDPs using tuples $\langle \mathcal{S}, \mathcal{A}, \mathcal{R}, \mathcal{T} \rangle$ where $\mathcal{S}$ is the set of states, $\mathcal{A}$ is the set of actions, $\mathcal{R} \colon \mathcal{S} \times \mathcal{A} \to \mathbb{R}$ is the reward function, and $\mathcal{T} \colon \mathcal{S} \times \mathcal{A} \to \Delta(\mathcal{S})$ is the transition function. An agent's interactions with an MDP are dictated by a policy $\pi \colon \mathcal{S} \to \Delta(\mathcal{A})$ mapping states to distributions over actions. We focus on episodic MDPs, meaning that after a finite number of transitions have occurred, the MDP will terminate. The history of states and actions is notated using $h = (s^0, a^0, \ldots, s^t)$. We use the notation $\mathcal{R}(h) = \sum_j \mathcal{R}(s^j, a^j)$ to denote the amount of reward accumulated over the course of a history. When a history is terminal, we use $z$ to notate it, rather than $h$. The objective of an MDP is to determine a policy $\arg\max_\pi \mathbb{E}_\pi \mathcal{R}(Z)$ yielding a large cumulative reward in expectation.

**Entropy** To help us quantify the idea of uncertainty, we introduce entropy. Symbolically, the entropy of a random variable $X$ is $\mathcal{H}(X) = -\mathbb{E} \log \mathcal{P}(X)$. Because the logarithm function is concave, the entropy of $X$ is maximized when the mass of $\mathcal{P}_X$ is spread as evenly as possible and minimized when the mass of $\mathcal{P}_X$ is concentrated at a single point.

In the context of decision-making, entropy can be used to describe the uncertainty regarding which action will be taken by an agent. When a policy spans multiple decision-points, the uncertainty regarding the agent's actions given that the state is known is naturally described by conditional entropy. Conditional entropy is the entropy of a random variable, conditioned upon the fact that the realization of another random variable is known. More formally, conditional entropy is defined by $\mathcal{H}(X \mid Y) = \mathcal{H}(X, Y) - \mathcal{H}(Y)$ where the joint entropy $\mathcal{H}(X, Y) = -\mathbb{E} \log \mathcal{P}(X, Y)$ is defined as the entropy of $(X, Y)$ considered as a random vector.

In some contexts, it is desirable for a decision-maker's policy to be highly stochastic. In such cases, an attractive alternative to the expected cumulative reward objective is the maximum entropy RL objective Ziebart et al. (2008) $\max_\pi \mathbb{E}_\pi \left[ \sum_t \mathcal{R}(S^t, A^t) + \alpha \mathcal{H}(A^t \mid S^t) \right]$, which trades off between maximizing expected return and pursuing trajectories along which its actions have large cumulative conditional entropy, using the temperature hyperparameter $\alpha$.

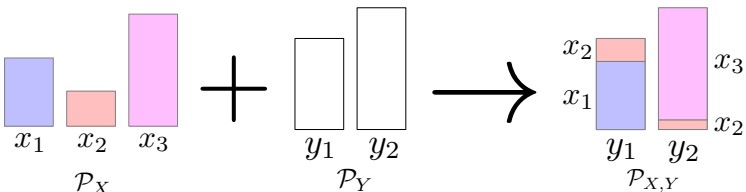

Figure 1: An approximate minimum entropy coupling. Given marginal distributions $\mathcal{P}_X$ and $\mathcal{P}_Y$, minimum entropy coupling constructs the a joint distribution $\mathcal{P}_{X,Y}$ having minimal joint entropy. In other words, it finds a joint distribution which allows to encode as much information as possible about $X$ into a given marginal distribution $\mathcal{P}_Y$.

**Mutual Information** A closely related concept to entropy is mutual information. Mutual information describes the strength of the dependence between two random variables. The greater the mutual information between two random variables, the more the outcome of one affects the conditional distribution of the other. Symbolically, mutual information is defined by $\mathcal{I}(X;Y) = \mathcal{H}(Y) - \mathcal{H}(Y \mid X) = \mathcal{H}(X) - \mathcal{H}(X \mid Y)$. From this definition, we see explicitly that the mutual information of two random variables can be interpreted as the amount of uncertainty about one that is eliminated by observing the realization of the other.

Mutual information is important for communication because we may only be able to share the realization of an auxiliary random variable, rather than that of the random variable of interest. In such cases, maximizing the amount of communicated information amounts to maximizing the mutual information between the auxiliary random variable and the random variable of interest.

**The Data Processing Inequality** The independence relationships among random variables play an important role in determining their mutual information. If random variables $X$ and $Z$ are conditionally independent given $Y$ (that is, $X \perp Z \mid Y$), the data processing inequality states that $\mathcal{I}(X;Y) \geq \mathcal{I}(X;Z)$. Less formally, the data processing inequality states that if $Z$ does not provide additional information about $X$ given $Y$, then the dependence between $X$ and $Z$ cannot be stronger than the dependence between $X$ and $Y$.

**Minimum Entropy Coupling** In some cases, we may wish to maximize the mutual information between two random variables subject to fixed marginals. That is, we are tasked with determining a joint distribution $\mathcal{P}_{X,Y}$ that maximizes the mutual information $\mathcal{I}(X;Y)$ between $X$ and $Y$ subject to the constraints that $\mathcal{P}_{X,Y}$ marginalizes to $\mathcal{P}_X$ and $\mathcal{P}_Y$, where $\mathcal{P}_X$ and $\mathcal{P}_Y$ are given as input. Invoking the relationship between mutual information and joint entropy $\mathcal{I}(X;Y) = \mathcal{H}(X) + \mathcal{H}(Y) - \mathcal{H}(X,Y)$, we see that this problem is equivalent to that of minimizing the joint entropy of $X$ and $Y$. As a result, this problem is often referred to as the minimum entropy coupling problem. A visual example is shown in Figure 1. While minimum entropy coupling is NP-hard Kovačević et al. (2015), Cicalese et al. (2019) recently showed that there exists a polynomial time algorithm that is suboptimal by no more than one bit.

## 4 MARKOV CODING GAMES

We are now ready to introduce Markov coding games (MCGs). An MCG is a tuple $\langle (\mathcal{S}, \mathcal{A}, \mathcal{T}, \mathcal{R}), \mathcal{M}, \mu, \zeta \rangle$, where $(\mathcal{S}, \mathcal{A}, \mathcal{T}, \mathcal{R})$ is a Markov decision process, $\mathcal{M}$ is a set of messages, $\mu$ is a distribution over $\mathcal{M}$, and $\zeta$ is a non-negative real number. An Markov coding game proceeds in the following steps:

1. First, a message $M \sim \mu$ is sampled from the prior over messages and revealed to the sender.
2. Second, the sender uses a message conditional policy to generate a trajectory $Z \sim (\mathcal{T}, \pi_{|M})$.
3. Third, the sender's terminal trajectory $Z$ is given to the receiver as an observation.
4. Fourth, the receiver uses a trajectory conditional policy to estimate the message $\hat{M} \sim \pi_{|Z}(Z)$.

The objective of the agents is to maximize the expected weighted sum of the return and the accuracy of the receiver's estimate $\mathbb{E}\left[\mathcal{R}(Z) + \zeta\mathbb{I}[M = \hat{M}]\right]$. Optionally, in cases in which a reasonable distance function is available, we allow for the objective to be modified to minimizing the distance

between the message and the guess $d(M, \hat{M})$, rather than maximizing the probability that the guess is correct.

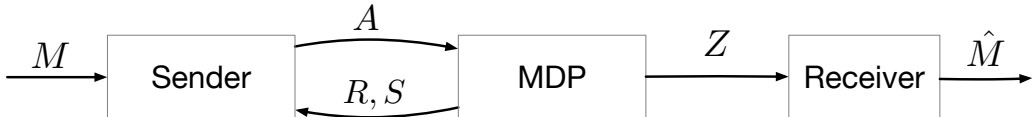

Figure 2: **A depiction of the structure of MCGs.** First, the sender is given a message. Second, the sender is tasked with a MDP, unrelated to the message. Third, the receiver observes the sender's trajectory. Fourth, the receiver estimates the message.

## 4.1 AN EXAMPLE

A payoff matrix for a simple MCG is shown in Figure 3. In this game, the sender is given one of two messages $m$ or $m'$, with equal probability. It then chooses among three actions $a_1$, $a_1'$ and $a_1''$, for which the rewards are 4, 3, and 0, respectively. The receiver observes the sender's trajectory (which is equivalent to the sender's action in MDPs with one state) and estimates the message using actions $\hat{m}$ and $\hat{m}'$, corresponding to guessing to $m$ and $m'$, respectively. The receiver accrues a reward of 4 for guessing correctly. The payoff entries in table denote $(\mathcal{R}(Z), \zeta \mathbb{I}[M = \hat{M}])$ for each outcome.

Figure 3: A payoff matrix for a simple MCG.

| Message | Sender | Receiver $\hat{m}$ | $\hat{m}'$ |
|---|---|---|---|
| $m$ | $a_1$ | $(4, \zeta)$ | $(4, 0)$ |
|  | $a_1'$ | $(3, \zeta)$ | $(3, 0)$ |
|  | $a_1''$ | $(0, \zeta)$ | $(0, 0)$ |
| $m'$ | $a_1$ | $(4, 0)$ | $(4, \zeta)$ |
|  | $a_1'$ | $(3, 0)$ | $(3, \zeta)$ |
|  | $a_1''$ | $(0, 0)$ | $(0, \zeta)$ |

As is generally true of MCGs, this MCG is difficult for independent approximate dynamic programming-based approaches because their learning dynamics are subject to local optima. Consider a run in which the sender first learns to maximize its immediate reward by always selecting $a_1$. Now, the receiver has no reason to condition its guess on the sender's action because the sender is not communicating any information about the message. As a result, thereafter, the sender has no incentive to communicate any information in its message, because the receiver would ignore it anyways. This outcome, sometimes called a babbling equilibrium, leads to a total expected return of $4 + \zeta/2$ (sender always selects $a_1$, receiver guesses randomly). In cases in which $\zeta$ is small (communication is unimportant), the babbling equilibrium performs well. However, it can perform arbitrarily poorly as $\zeta$ becomes large.

## 4.2 SPECIAL CASES

We can express both (a large class of) referential games and various source coding settings as special cases of the MCG formalism by describing the MDPs to which they correspond.

**(A Large Class of) Referential Games** We can express a $T$ step referential game as follows.

- The state space $\mathcal{S} = \{s^0, s^1, \ldots, s^T\}$.
- The transition function deterministically maps $s^t \mapsto s^{t+1}$ and terminates at input $s^T$.
- The reward function maps to zero for every state action pair.

**Standard Source Coding** We can express the standard source coding problem as follows.

- The state space $\mathcal{S} = \{s\}$.
- The action space $\mathcal{A} = \tilde{\mathcal{A}} \cup \{\emptyset\}$.
- The transition function deterministically maps to $s$ to $s$ for $a \in \tilde{A}$ and terminates the game on $\emptyset$.
- The reward function maps to $-1$ for $a \in \tilde{\mathcal{A}}$ and maps $\emptyset$ to 0.

**Length Limited Source Coding** Length limited source coding can be captured in the same way as standard source coding with the modifications that $\mathcal{S} = \{s^0, s^1, \ldots, s^T\}$, the transition function terminates on $s^T$, and the reward function yields 0 from $s^T$, where $T$ is the length limit.

**Source Coding with Unequal Symbol Costs** Source coding with unequal symbol costs can be captured in the same way as standard source coding with the modification that $\mathcal{R}(\cdot, a)$ returns the negative symbol cost of $a$, rather than returning $-1$, for $a \in \tilde{\mathcal{A}}$.

## 5 GREEDY MINIMUM ENTROPY COUPLING

To address MCGs, we propose a novel algorithm we call greedy minimum entropy coupling (GME). GME (and more broadly, any algorithm geared toward MCGs) is faced with two competing incentives. On one hand, it needs to maximize expected reward $\mathcal{R}(Z)$ generated by the MDP. On the other hand, it needs to maximize the amount of information $\mathcal{I}(M; Z)$ communicated to the receiver about the message, so as to maximize the probability of a correct guess. GME handles this trade-off using a two step process for constructing the sender's policy. In the first step, it computes an MDP policy that balances between high cumulative reward and large cumulative entropy. In the second step, it couples the probability mass of this policy with the posterior over messages in such a way that the expected return does not decrease and the amount of mutual information between the message and trajectory is greedily maximized. We describe these steps below. Thereafter, we show this procedure possesses desirable guarantees and discuss intuition for the method.

### 5.1 METHOD DESCRIPTION

**Step One** In the first step, GME uses MaxEnt RL to construct an MDP policy $\pi$. This policy is an MDP policy, not an MCG policy, and therefore does not depend on the message. Note that this policy depends on the choice of temperature $\alpha$ used for the MaxEnt RL algorithm.

**Step Two** In the second step, at execution time, GME constructs a message-conditional policy online. Say that, up to time $t$, the sender is in state $s^t$, history $h^t$ and has played according to the message conditional policy $\pi_{|M}^{:t}$. Let

$$b^t = \mathcal{P}(M \mid h^t, \pi_{|M}^{:t})$$

be the posterior over the message, conditioned on the history and the historical policy. GME performs a MEC between the posterior over the message $b^t$ and distribution over actions $\pi(s^t)$, as determined by the MDP policy. Let $\nu = \text{MEC}(b^t, \pi(s^t))$ denote joint distribution over messages and actions resulting from the coupling. Then GME sets the sender to act according to the message conditional distribution

$$\pi_{|M}^t(s^t, m) = \nu(A^t \mid M = m)$$

of the coupling distribution $\text{MEC}(b^t, \pi(s^t))$.

Given the sender's MDP trajectory, GME's receiver uses the the sender's MDP policy and MEC procedure to reconstruct the sender's message conditional policy along the trajectory; thereafter, the receiver computes the posterior and guesses the maximum a posteriori message.

Pseudocode for GME is included in the appendix.

### 5.2 METHOD ANALYSIS

GME possesses guarantees both concerning the return $\mathcal{R}(Z)$ generated by the MDP and concerning the amount of information communicated $\mathcal{I}(M; Z)$.

**Proposition 1.** *At each state of the MDP, the message conditional policy $\pi_{|M}$ selects actions with the same probabilities as the MDP policy $\pi$.*

*Proof.* Fix an arbitrary state $s$. Let $b$ be a posterior over the message induced by the sender's message conditional policy on the way to $s$. Then recall that GME uses the distribution $\mathcal{P}_{\text{MEC}(b,\pi(s))}$ induced by a MEC between $b$ and $\pi(s)$ to select its action. Because MECs guarantee that the resultant joint distribution marginalizes correctly, it follows directly that GME must select its actions with the same probabilities as $\pi(s)$. $\square$

**Proposition 2.** *The expected return generated from the MDP by the message conditional policy $\pi_{|M}$ is equal to that of the MaxEnt policy $\pi$. That is,*

$$\mathbb{E}_{M \sim \mu} \mathbb{E}_{\pi_{|M}} \mathcal{R}(Z) = \mathbb{E}_\pi \mathcal{R}(Z).$$

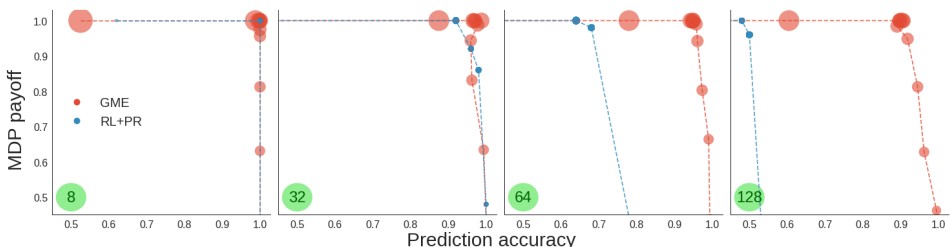

Figure 4: Results for GME and RL+PR on CodeGrid with varying message space sizes.

*Proof.* It follows from Proposition 1 that all trajectories are generated with the same probabilities and therefore that the expected returns are the same. □

**Proposition 3.** *At each decision point, GME greedily maximizes the mutual information $I(M; H^{t+1} \mid b^t, h^t)$ between the message $M$ the history at the next time step $H^{t+1}$, given the contemporaneous posterior and history, subject to Proposition 1.*

*Proof.* For conciseness, we leave the conditional dependence on $b^t$ and $h^t$ implicit in the argument. First, we claim that $\mathcal{I}(M; H^{t+1}) = \mathcal{I}(M; A^t)$. To see this, first consider that $H^{t+1} \equiv (h^t, A^t, S^{t+1})$. This means we have $\mathcal{I}(M; H^{t+1}) = \mathcal{I}(M; (A^t, S^{t+1}))$ since we are conditioning on $h^t$. Now, consider that because the message influences the next state only by means of the action, we have the causal graph $M \to A^t \to (A^t, S^{t+1})$, which implies that $M \perp (A^t, S^{t+1}) \mid A^t$. Also, we trivially have $M \perp A^t \mid (A^t, S^{t+1})$. These conditional independence properties allow us to apply the data processing inequality: $X \perp Z \mid Y \Rightarrow \mathcal{I}(X; Y) \geq \mathcal{I}(X; Z)$. Applying it in both directions yields $\mathcal{I}(M; (A^t, S^{t+1})) = \mathcal{I}(M; A^t)$.

Now consider that we can rewrite mutual information using the equality

$$\mathcal{I}(M; A^t) = \mathcal{H}(M) + \mathcal{H}(A^t) - \mathcal{H}(M, A^t). \tag{1}$$

The first term $\mathcal{H}(M)$ is exogenous by virtue of being determined by $b^t$. The second term is exogenous by virtue of being subject to Proposition 1. The third term is the joint entropy between $M$ and $A^t$, which is exactly what a minimum entropy coupling minimizes. □

### 5.3 METHOD DISCUSSION

One aspect of GME that went uncommented upon in the analysis section is the choice of MaxEnt RL for step one. The reason that GME uses MaxEnt RL here is to control the $\mathcal{H}(A^t)$ term in equation (1). If the temperature $\alpha$ is large, GME places more emphasis on maximizing $\mathcal{H}(A^t)$ (and thereby $\mathcal{I}(M; Z)$); if the temperature $\alpha$ is small, GME places more emphasis on maximizing $\mathcal{R}(Z)$. The appropriate choice of $\alpha$ will depend on the value of $\zeta$. Code for GME that illustrates the functionality of $\alpha$ applied to the example from Figure 3 can be found at `https://bit.ly/36I3LDm`.

A second aspect of GME that we have yet to discuss is its scalability. Performing an approximate MEC takes $O(N \log N)$ time, making it expensive to scale to very large message spaces. To accommodate this fact, for large message spaces we recommend using a factored representation $\mathbb{M} \subset \mathbb{M}_1 \times \cdots \times \mathbb{M}_k$ and a corresponding factored belief $(b_1, \ldots, b_k)$, where each $b_j$ tracks the posterior over $\mathbb{M}_j$. At each time step, we suggest performing a MEC between $\pi(s^t)$ and the block $b_j = \arg\max_{b_j} \mathcal{H}(b_j)$ having the largest entropy. By doing so, we can substantially reduce the size the of the space over which we need to perform a minimum entropy coupling at any one time, allowing us to scale to extremely large message spaces.

## 6 EXPERIMENTS

We investigate the efficacy of GME on MCGs based on a gridworld, Cartpole and Pong. For all three we use MaxEnt Q-learning for our MDP policy. Details regarding our implementation can be found in the supplementary material and in the appendix.

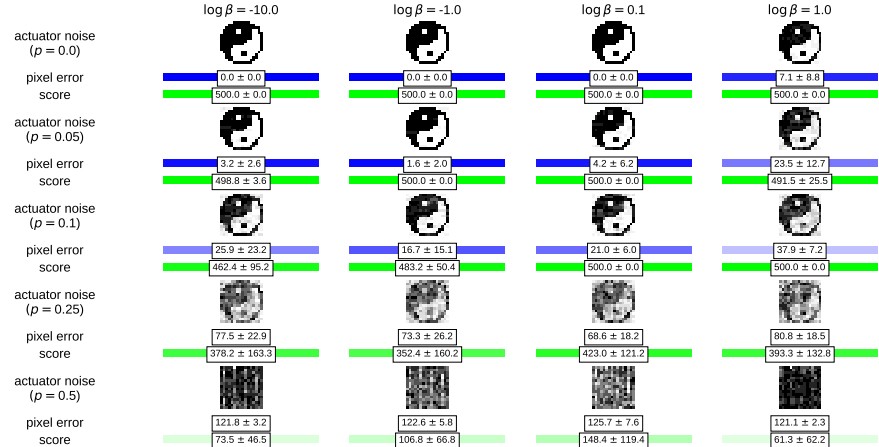

Figure 5: Results for GME on CodeCart with varying amounts of actuator noise and temperatures.

**CodeGrid** In our gridworld MCG, which we call CodeGrid, the sender is placed on a $4 \times 4$ grid in which it can move left, right, up and down. The sender starts at position $(1, 1)$ and must to move to $(4, 4)$ within 8 time steps to achieve the maximal MDP return of 1. Otherwise, the game terminates after 8 time steps and the sender receives a payoff of 0.

For our baseline, we trained an RL agent to play as the MCG sender. We paired this RL agent with an perfect receiver, meaning that, for each episode, the receiver computed the exact posterior over the message based on the sender's current policy and guessed the MAP, both during training and testing. We abbreviate this baseline as RL+PR (where PR stands for perfect receiver). RL+PR is one way to adapt related work such as (Strouse et al., 2018; Eysenbach et al., 2019) to the MCG setting. Pseudocode for the RL+PR baseline can be found in the appendix.

We show results for GME and RL+PR across a variety of settings in Figure 7. The column of the figure corresponds to the cardinality of the message space; the exact size is listed in the green bubble. We use a uniform marginal over messages in each case. The $x$-axis corresponds to the proportion of the time that the receiver correctly guesses the message. The $y$-axis corresponds to the MDP payoff (in this case whether the sender reached the opposing corner of the grid within the time limit). Both GME and RL+PR possess mechanisms to trade-off between these two goals. GME can adjust its temperature $\alpha$, while RL+PR can adjust the value of $\zeta$ used during training. Figure 7 show the Pareto curve for each. For GME, the size of the circle corresponds to the inverse temperature $\beta = 1/\alpha$.

For the settings with 8 messages and 32 messages, we observe that both GME and RL+PR achieve strong performance—achieving optimal or nearly optimal MDP returns and optimal or nearly optimal transmission. However, as the size of the message space increases, the performance of RL+PR falls off sharply. Indeed, for the 128 message setting, RL+PR is only able to correctly transmit the message slightly more than half the time. RL+PR's inability to achieve strong performance in these cases may be a result of the fact that communication protocols violate the locality assumption of approximate dynamic programming approaches. In contrast, we observe that GME, which constructs its protocol using MEC, remains optimal or near optimal both the 64 and the 128 message settings.

**CodeCart and CodePong** While the CodeGrid experiments that GME can outperform an obvious approach to MCGs, it remains to be determined whether GME can perform well at scale. Toward the end of making this determination, we consider MCGs based on Cartpole and Pong. In these MCGs, the message spaces are the sets of $16 \times 16$ and $22 \times 22$ binary images, respectively, each with a uniform prior. The cardinality of these spaces ($> 10^{77}$ and $> 10^{145}$) is astronomical. In fact, it is not clear how an RL+PR-like approach could even be scaled to this setting. On the other hand, GME is easily adaptable to this setting, using the factorization scheme suggested in the method section.

We also include results with variable amounts of *actuator noise*, i.e., with some probability $p$, the environment executes a random action, rather than the one it intended. Actuator noise models the probability of error during transmission and arises naturally in many settings involving communication, such as noisy channel coding. In this setting, the receiver may only observe the action executed

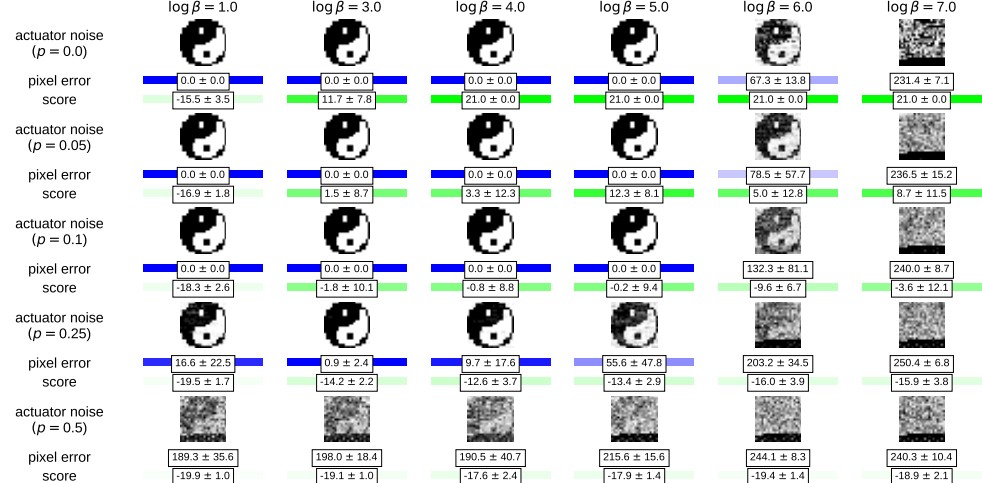

Figure 6: Results for GME on CodePong with varying amounts of actuator noise and temperatures.

by the environment, rather than the one intended by the sender. While this setting is not technically an MCG, GME can still be directly applied. And while the derivation of Proposition 3 no longer holds, we suggest that GME nevertheless offers a strong heuristic in such settings.

We show the results in Figure 5 and Figure 6 for CodeCart and CodePong, respectively. For both plots, the $y$-axis corresponds to the amount of actuator noise (lower is more noise). The $x$-axis corresponds the inverse temperature value $\beta = 1/\alpha$ (further right is colder temperature, meaning there is more emphasis on MDP expected return). Each entry contains a decoded yin-yang symbol with the corresponding temperature and actuator noise. Each entry also lists the $\ell_1$ pixel error (blue) and the MDP expected return (green), along with corresponding standard errors over 10 runs; a brighter color corresponds to better performance.

Remarkably, for both CodeCart and CodePong, we observe that, when there is no actuator noise, GME is able to perfectly transmit images while achieving the maximum expected return in the MDP. For CodeCart, this occurs at $\log \beta \in \{-10, -1, 0.1\}$; for CodePong, it occurs at $\log \beta \in \{4, 5\}$. Interestingly, as the amount of actuator noise increases to a non-zero value, the effect on performance differs between CodeCart and CodePong. In CodeCart, GME continues to achieve maximal performance in the MDP, but begins to accumulate errors in the transmission. In contrast, in CodePong, GME continues to transmit the message with perfect fidelity, but begins to lose expected return from the MDP. This suggests that accidentally taking a random action is costlier in Pong than it is in Cartpole, but that, in an informal sense, Pong has more bandwidth to transmit information. However, in both cases, the decay in performance is graceful—for example, with $p = 0.05$, the decrease in the visual quality of the transmission for Cartpole ($\log \beta = -1$) is difficult to even perceive, while the CodePong sender still manages to win games roughly 80% of the time ($\log \beta = 5$). The performance in both settings continues to deteriorate up to $p = 1/2$, at which point GME is neither able to perform adequately on the MDP nor to transmit a clear symbol.

**Videos of the agent playing are included in the supplemental material.**

## 7 CONCLUSIONS

This work introduces a new problem setting called Markov coding games, which generalize referential games and source coding and are related to channel coding problems and decentralized control. We contribute an algorithm for this setting called GME and show that GME possesses provable guarantees regarding both the return generated from the MDP and the amount of information content communicated, subject to some constraints. On the experimental side, we show that GME significantly outperforms an RL baseline with a perfect receiver. Finally, we show that GME is able to scale to extremely large message spaces and transmit these messages with high fidelity, even with some actuator noise, suggesting that it could be robust in real world settings.

## 8 REPRODUCIBILITY

Our codebase is included in the supplemental material.

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

## A  EXPERIMENTAL DETAILS

For CodeGrid, we use a policy parameterized by neural network with two fully-connected layers of hidden dimension 64, each followed by a ReLu activation (Nair & Hinton, 2010). For CodePong and CodeCart, we use a convolutional encoder with three layers of convolutions (number of channels, kernel size, stride) as follows: (32,8,4), (64,4,2), (64,3,1). This is followed by a fully connected layer (Mnih et al., 2015). We use ReLu activations after each layer, note that we do not use any max-pooling. For CodeGrid and CodePong, layer weights are randomly initialized using PyTorch 1.7 (Paszke et al., 2017) defaults. For CodeCart, we initialise weights according to an optimally-trained DQN policy included in *rl-baselines3-zoo*[1] and individually finetune for each $\beta$ for $15k$ steps. For all environments, we use the Adam optimizer with learning rate $10^{-4}$, $\beta_1 = 0.9$, $\beta_2 = 0.999$, $\epsilon = 10^{-8}$ and no weight decay.

**CodeGrid**

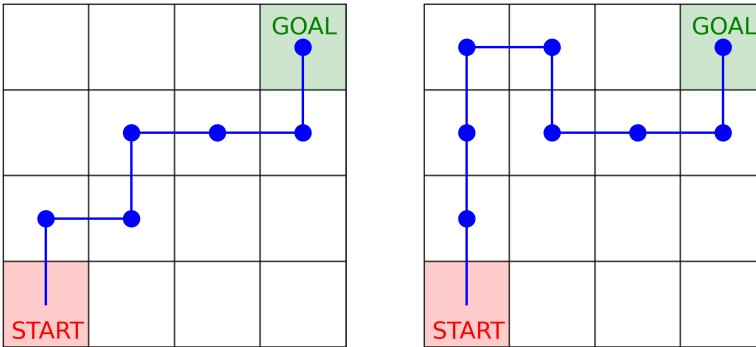

Figure 7: An illustration of two possible CodeGrid trajectories, one of length 6 (left) and one of length 8 (right).

**CodeCart**  For CodeCart we use the standard action space (move left and move right). There are no redundant actions that the agent can use to communicate.

**CodePong**  For CodePong we use a reduced action space of size two (move up and move down). There are no redundant actions that the agent can use to communicate.

---

[1] https://github.com/DLR-RM/rl-baselines3-zoo

# B ALGORITHMIC DETAILS

## B.1 GME

---

**Algorithm 1** GME

---
// Step 1: Compute MDP policy
**Input:** MDP $G_{\text{MDP}}$, temperature $\alpha$
MaxEnt MDP policy $\pi \leftarrow \text{MaxEntRL}(G_{\text{MDP}}, \alpha)$

// Step 2: Play Sender's Part of MCG episode
**Input:** MDP policy $\pi$, MCG $G_{\text{MCG}}$
message $m \leftarrow \text{reset}(G_{\text{MCG}})$ // observe message from environment
belief $b \leftarrow G_{\text{MCG}}.\mu$
state $s \leftarrow G_{\text{MCG}}.s^0$
**while** sender's turn **do**
   joint distribution $\nu \leftarrow \text{minimum\_entropy\_coupling}(b, \pi(s))$
   decision rule $\pi_{|M} \leftarrow \nu(A \mid M)$
   sender action $a \sim \pi_{|M}(m)$
   new belief $b \leftarrow \text{posterior\_update}(b, \pi_{|M}, a)$
   next state $s \leftarrow G_{\text{MCG}}.\text{step}(a)$
**end while**

// Step 3: Play Receiver's Part of MCG episode
**Input:** MDP policy $\pi$, MCG $G_{\text{MCG}}$
MDP trajectory $z \leftarrow G_{\text{MCG}}.\text{receiver\_observation}()$
belief $b \leftarrow G_{\text{MCG}}.\mu$
**for** $s, a \in z$ **do**
   joint distribution $\nu \leftarrow \text{minimum\_entropy\_coupling}(b, \pi(s))$
   decision rule $\pi_{|M} \leftarrow \nu(A \mid M)$
   new belief $b \leftarrow \text{posterior\_update}(b, \pi_{|M}, a)$
**end for**
estimated message $\hat{m} \leftarrow \arg\max_{m'} b(m')$
$G_{\text{MCG}}.\text{step}(\hat{m})$

---

## B.2 FACTORED GME

---
**Algorithm 2** Factored GME
---
    // Step 1: Compute MDP policy
    **Input:** MDP $G_{\text{MDP}}$, temperature $\alpha$
    MaxEnt MDP policy $\pi \leftarrow \text{MaxEntRL}(G_{\text{MDP}}, \alpha)$

    // Step 2: Play Sender's Part of MCG episode
    **Input:** MDP policy $\pi$, MCG $G_{\text{MCG}}$
    message $m \leftarrow \text{reset}(G_{\text{MCG}})$ // observe message from environment
    belief $b \leftarrow G_{\text{MCG}}.\mu$
    state $s \leftarrow G_{\text{MCG}}.s^0$
    **while** sender's turn **do**
        active block index $i \leftarrow \arg\max_j \{\mathcal{H}(b_j)|b_j \in b\}$
        joint distribution $\nu \leftarrow \text{minimum\_entropy\_coupling}(b_i, \pi(s))$
        decision rule $\pi_{|M} \leftarrow \nu(A \mid M)$
        sender action $a \sim \pi_{|M}(m_i)$
        new belief $b_i \leftarrow \text{posterior\_update}(b_i, \pi_{|M}, a)$
        next state $s \leftarrow G_{\text{MCG}}.\text{step}(a)$
    **end while**

    // Step 3: Play Receiver's Part of MCG episode
    **Input:** MDP policy $\pi$, MCG $G_{\text{MCG}}$
    MDP trajectory $z \leftarrow G_{\text{MCG}}.\text{receiver\_observation}()$
    belief $b \leftarrow G_{\text{MCG}}.\mu$
    **for** $s, a \in z$ **do**
        active block index $i \leftarrow \arg\max_j \{\mathcal{H}(b_j)|b_j \in b\}$
        joint distribution $\nu \leftarrow \text{minimum\_entropy\_coupling}(b_i, \pi(s))$
        decision rule $\pi_{|M} \leftarrow \nu(A \mid M)$
        new belief $b \leftarrow \text{posterior\_update}(b, \pi_{|M}, a)$
    **end for**
    **for** $i$ **do**
        estimated $i$th message block $\hat{m}_i \leftarrow \arg\max_{m'_i} b_i(m'_i)$
    **end for**
    $G_{\text{MCG}}.\text{step}(\hat{m})$
---

## B.3 RL+PR BASELINE

---
**Algorithm 3** RL+PR baseline
---
    // Play Episode
    **Input:** MCG $G_{\text{MCG}}$
    state $s \leftarrow s^0$
    message $m \leftarrow \text{sample}(\mu)$
    belief $b \leftarrow \mu$
    **while** True **do**
        action $a \leftarrow \text{sample}(\pi(s, m))$
        belief $b \leftarrow \text{posterior\_update}(b, \pi, a)$
        new state $s' \leftarrow \text{sample}(\mathcal{T}(s, a))$
        **if** $s'$ non-terminal **then**
            add\_to\_buffer$(s, a, \mathcal{R}(s, a), s')$
            state $s \leftarrow s'$
        **else**
            break
        **end if**
    **end while**
    add\_to\_buffer$(s, a, \mathcal{R}(s, a) + \zeta \max_{m' \in \mathcal{M}} b(m'), \emptyset)$
---

### B.4 MIN ENTROPY JOINT DISTRIBUTION ALGORITHM OUTPUTTING A SPARSE REPRESENTATION OF $M$ CICALESE ET AL. (2019)

---

**Algorithm 4** Min Entropy Joint Distribution

---

**Require:** prob. distributions $\mathbf{p} = (p_1, \ldots, p_n)$ and $\mathbf{q} = (q_1, \ldots, q_n)$
**Ensure:** A Coupling $\mathbf{M} = [m_{ij}]$ of $\mathbf{p}$ and $\mathbf{q}$ in sparse representation $\mathbf{L} = \{(m_{ij}, (i, j)) \,|\, m_{ij} \neq 0\}$

  **if** $\mathbf{p} \neq \mathbf{q}$, let $i = \max\{j | p_j \neq q_j\}$; **if** $p_i < q_i$ **then swap** $\mathbf{p} \leftrightarrow \mathbf{q}$.
  $\mathbf{z} = (z_1, \ldots, z_n) \leftarrow \mathbf{p} \wedge \mathbf{q}, \mathbf{L} \leftarrow \emptyset$
  CreatePriorityQueue$\left(\mathcal{Q}^{(row)}\right)$, qrowsum $\leftarrow 0$
  CreatePriorityQueue$\left(\mathcal{Q}^{(col)}\right)$, qcolsum $\leftarrow 0$
  **for** $i = n$ **downto** 1 **do**
    $z_i^{(d)} \leftarrow z_i, \; z_i^{(r)} \leftarrow 0$
    **if** $qcolsum + z_i > q_i$ **then**
      $\left(z_i^{(d)}, z_i^{(r)}, I, qcolsum\right) \leftarrow$ Lemma3-Sparse $\left(z_i, q_i, \mathcal{Q}^{(col)}, qcolsum\right)$
    **else**
      **while** $\mathcal{Q}^{(col)} \neq \emptyset$ **do**
        $(m, l) \leftarrow$ ExtractMin$(\mathcal{Q}^{(col)})$,
        $qcolsum \leftarrow qcolsum - m$,
        $\mathbf{L} \leftarrow \mathbf{L} \cup \{(m, (l, i))\}$
      **end while**
    **end if**
    **if** $qrowsum + z_i > p_i$ **then**
      $(z_i^{(d)}, z_i^{(r)}, I, qrowsum) \leftarrow$ Lemma3-Sparse$(z_i, p_i, \mathcal{Q}^{(row)}, qrowsum)$
      **for each** $(m, l) \in I$ **do** $\mathbf{L} \to \mathbf{L} \cup \{(m, (i, l))\}$
      **if** $z_i^{(r)} > 0$ **then** Insert $\left(\mathcal{Q}^{(row)}, (z_i^{(r)}, i)\right)$
      $qrowsum \leftarrow qrowsum + z_i^{(r)}$
    **else**
      **while** $\mathcal{Q}^{(row)} \neq \emptyset$ **do**
        $(m, l) \leftarrow$ ExtractMin$(\mathcal{Q}^{(row)})$
        $qrowsum \leftarrow qrowsum - m$
        $\mathbf{L} \leftarrow \mathbf{L} \cup \{(m, (i, l))\}$
      **end while**
    **end if**
    $\mathbf{L} \leftarrow \mathbf{L} \cup \left\{(z_i^{(d)}, (i, i))\right\}$
  **end for**

---

**Algorithm 5** Lemma3-Sparse

---

**Require:** real $z > 0$, $x \geq 0$, and priority queue $\mathcal{Q}$ s.t. $\left(\sum_{(m,l) \in \mathcal{Q}} m\right) = qsum$ and $qsum = x \geq z$
**Ensure:** $z^{(d)}, z^{(r)} \geq 0$, and $I \subseteq \mathcal{Q}$ s.t. $z^{(d)} + z^{(r)} = z$, and $z^{(d)} + \sum_{(m,l) \in I} m = x$
  $I \leftarrow \emptyset, \; sum \leftarrow 0$
  **while** $\mathcal{Q} \neq \emptyset$ **and** $sum + Min(\mathcal{Q}) < x$ **do**
    $(m, l) \leftarrow$ ExtractMin$(\mathcal{Q}), \; qsum \leftarrow qsum - m$
    $I \leftarrow I \cup \{(m, l)\}, \; z^{(r)} \leftarrow z - z^{(d)}$
  **end while**
  $z^{(d)} \leftarrow x - sum, \; z^{(r)} \leftarrow z - z^{(d)}$
  **return** $\left(z^{(d)}, z^{(r)}, I, qsum\right)$

---

