# OpenReview forum: "Communicating via Markov Decision Processes"
_ICLR.cc/2022/Conference — ICLR 2022 Submitted_

### Official Review · Reviewer_8Cj5 · 2021-10-29

**Correctness:** 2
**Technical Novelty And Significance:** 2
**Empirical Novelty And Significance:** 2
**Recommendation:** 3
**Confidence:** 4

**Main Review:**

Pro:
- Using MDP policies to code messages is a fun and interesting idea.
- The Decentralized POMDP framework seems a natural formalism for this problem
- Introducing a stylized theoretical game to formalize a difficult point in a more complex problem (like for instance multi-armed bandits for exploration/exploitation in RL) can provide valuable insights

Cons:
- The key point when introducing new theoretical games is to give a clear, abstract view of a problem. But I found the formalization of this MCG problem too muddled. To start with, it is not clear to me if we are really facing a two-players game or a single-player (the sender) game because the way the receiver "guesses" the message is not clearly formalized in the paper.

Questions:
- Assuming the game is a single-player game, is it reducible to an equivalent "transformed" MDP/POMDP where original MDP reward is shaped to reflect the additional "easy-to-guess" reward ?
- If we are facing a two-players game, is it a special case of Markov game as in (Littman 94) ?
- Is the proposed GME algorithm able to solve a special case of MCG, as for instance a channel coding problem as efficiently as an ad-hoc algorithm would ?


Minor remarks:
p4 "An Markov" -> "A Markov"
p4 "the receiver uses a trajectory conditional policy to guess the message \hat{M} \sim \pi_{|Z}(Z)"  is not a proper formalization.




**Summary Of The Paper:**

The article introduce a new theoretical game called Markov Coding Game (MCG).
A Markov Coding game is a special case of Decentralized POMDP (Oliehoek et al. 2016). It seems to be a two-players cooperative game with a sender agent and a receiver agent. Given a message unknown to the receiver, the sender agent must play on a fixed MDP in a way that facilitates the decoding of this message by the receiver.
A dedicated algorithm called GME for greedy minimum entropy coupling is proposed to solve this problem.
According to the authors, this game is supposed to generalize several referential games and channel coding games.
Some experiments are provided where bitmap images are transmitted through actions on gridword and pong environments.



**Summary Of The Review:**

A new theoretical game called Markov Coding Game and an algorithm to solve it. But I found the formalization the problem too muddled.

---

> ### Author Response · Authors · 2021-11-16
> **Thanks for the review!**
>
> > the way the receiver "guesses" the message is not clearly formalized in the paper.
>
> The sentence in the text “the receiver uses a trajectory conditional policy to guess the message \hat{M} \sim \pi_{|Z}(Z)" can be explained more elaborately as follows. The receiver observes the sender’s terminal MDP trajectory as its observation Z=(S^1, A^1, …, S^T, A^T). The receiver’s action space is the space of all messages \mathbb{M}. The receiver only acts once in the game -- after the sender’s MDP trajectory has terminated. After the receiver acts, the game ends.
>
> We agree that this point is crucial to communicate clearly. We would appreciate any feedback from the reviewer as to whether the text above clarifies the confusion, or whether there is additional information we can add to make this point more clear.
>
> > I found the formalization of this MCG problem too muddled
>
> Please let us know whether any parts of the problem formulation came across as unclear outside of the way the receiver “guesses” discussed above.
>
> > it is not clear to me if we are really facing a two-players game or a single-player
>
> The game is really a two-player game (one “sender” and one “receiver”). It is not a special case of a Markov game because the message is not observed by the receiver. MCGs can be viewed as Dec-POMDPs, POSGs, or extensive-form games.
>
> > is it reducible to an equivalent "transformed" MDP/POMDP where original MDP reward is shaped to reflect the additional "easy-to-guess" reward
>
> In general, it is possible to transform any Dec-POMDP to an equivalent POMDP (though this reduction is exponential). See Decentralized Stochastic Control with Partial History Sharing: A Common Information Approach (Nayyar 2012) or Optimally Solving Dec-POMDPs as Continuous-State MDPs (Dibangoye 2016). However, the problem setting we present here is “no more” transformable to a POMDP than any other multi-player common payoff game.
>
> > Is the proposed GME algorithm able to solve a special case of MCG, as for instance a channel coding problem as efficiently as an ad-hoc algorithm would ?
>
> To clarify, it is source coding, not channel coding, that is a special case of MCGs. We are not sure what the reviewer means by an “ad-hoc algorithm” here, but happy to discuss further if the reviewer wouldn’t mind clarifying.

---

### Official Review · Reviewer_FcQX · 2021-11-05

**Correctness:** 3
**Technical Novelty And Significance:** 2
**Empirical Novelty And Significance:** 2
**Recommendation:** 5
**Confidence:** 3

**Main Review:**

Strength:

1. The paper identifies some interesting connections of MCG with several existing problems.
2. The paper conducted extensive empirical evaluation of its model. The results look good and are presented with carefully-designed figures and videos. The performance comparison with RL+PR baseline is also interesting,

Weakness:

1. I am sorry to say the paper does not have sufficient novelty in both its proposed problem setting and the learning algorithm. First, I think the described problem setting, at least something very similar, already exists (see the reference below). The only difference might be that they do not explicitly evaluate on how accurate the message is decoded. But I am not sure why it is necessary to explicitly evaluate this objective, if we could design a game where good communication is necessary to achieve optimality. I hope the author can provide more motivation on the design of this problem setting. Second, technical contribution is limited, as the proposed GME algorithm seems to be a straightforward combination of existing methods, MaxEnt RL and MEC.
2. I am not sure why the authors describe their proposed method as "theoretically grounded", as I see no theoretical guarantee about its proposed method in this paper from the three Propositions in practice. I think it might be possible to combine proof techniques in information signaling scheme design (e.g., persuasiveness constraint) to improve the theoretical results.
3. In the empirical evaluation, the results look good, but I wish to see more baselines for comparisons. However, some more thorough tests are possible to help us understand the strength of the proposed method (e.g., See question 3 below).

Questions:

1. I wonder why the authors describe their problem setting as "the receiver observes the sender’s MDP trajectory", but from the examples, it actually just observes the sender's taken action. Is it just a fancy word, or the author is actually considering something more general about the receiver's observation.
2. I wonder how the proposed problem is the related to a series of literature in learning multiagent communication, and can their method serves as the additional baseline of this paper:
    - Mordatch, Igor, and Pieter Abbeel. "Emergence of grounded compositional language in multi-agent populations." Thirty-second AAAI conference on artificial intelligence. 2018.
    - S. Sukhbaatar, R. Fergus, et al. Learning multiagent communication with backpropagation. In
Advances in Neural Information Processing Systems, pages 2244–2252, 2016.
    - Lowe, Ryan, et al. "Multi-agent actor-critic for mixed cooperative-competitive environments." Proceedings of the 31st International Conference on Neural Information Processing Systems. 2017.
3. I notice that the games in the experiment do not include the additional dummy action available for communication, so the agent has to take the action that not only serves as to achieve value but also to achieve good communication with the receiver. I wonder what happens if the game includes the redundant action that the agent can use to communicate.



**Summary Of The Paper:**

The paper introduces a problem setting, namely Markov Coding Game (MCG): sender receives a state (message) and takes an action to communicate the state to the receiver, receiver observes the taken action, then takes an action to decode the observed state of sender. This game is related to the referential games, source coding as well as decentralized control. To solve the MCG problem, the author designs the greedy minimum entropy coupling algorithm (GME), that aims to maximize the returns of the MDP and learn a good communication protocol simultaneously by combining existing techniques, MaxEnt RL and MEC. The algorithm is test empricially on Gridworld, Cartpole, and Pong.


**Summary Of The Review:**

While the paper demonstrates some nice empirical results, my major concern is its lack of novelty in the proposed problem setting and the learning algorithm. Therefore, I think additional studies are needed in order to  improve the technical and empirical contribution of the work.

---

> ### Author Response · Authors · 2021-11-16
> **Thanks for the review!**
>
> We thank the reviewer for their review.
>
> > the described problem setting, at least something very similar, already exists (see the reference below)
>
> Is the reviewer referring to one of the references listed in question 2? We are not aware of this problem setting already existing, but would certainly like to know if it does. We would appreciate it if the reviewer could clarify the exact problem setting to which they are referring.
>
>
> > technical contribution is limited, as the proposed GME algorithm seems to be a straightforward combination of existing methods, MaxEnt RL and MEC.
>
> We view the fact that GME can be implemented as a straightforward combination of MaxEnt RL and MEC as a strength, not a weakness. Furthermore, we do not think this connection was necessarily obvious. MEC is far from a standard tool---it was not until 2019 that efficient approximation algorithms even existed. As far as we are aware, prior to this submission, MEC had never been used in the context of reinforcement learning. (The paper which derived the efficient approximation algorithm has only 10 citations at the time of writing!)
>
> > I am not sure why the authors describe their proposed method as "theoretically grounded"
>
> We used the language “theoretically grounded” to refer to Proposition 2 and Proposition 3. Proposition 2 does hold in practice and Proposition 3 holds up to the amount of approximation error of the MEC algorithm. Proposition 2 provides a guarantee about the cumulative reward generated from the MDP, while Proposition 3 guarantees that the amount of information communicated is myopically maximized, subject to Proposition 1.
>
> > I wonder why the authors describe their problem setting as "the receiver observes the sender’s MDP trajectory", but from the examples, it actually just observes the sender's taken action. Is it just a fancy word, or the author is actually considering something more general about the receiver's observation.
>
> We do really mean the trajectory. In our toy example, the trajectory is equivalent to the sender’s action because the MDP only has one state and only lasts one time step. However, in general (e.g. in our full scale experiments), the trajectory and action are not equivalent. That the receiver observes the trajectory, not just the actions, of the sender is crucial to both GME’s theoretical guarantees and practical performance.
>
> > I wonder how the proposed problem is the related to a series of literature in learning multiagent communication
>
> The Mordatch and Sukhbaatar papers are closely related to the Foerster paper discussed in the referential games section related work in that they learn communication using end-to-end differentiation. We will add them to our related work. We do not see the Lowe paper as particularly closely related beyond the fact that it also concerns cooperative multi-agent settings. Our current submission contains a multi-agent learning baseline with an *optimal receiver*, which we believe likely upper bounds the performance of standard CTDE MARL approaches. For completeness we’ll add the weaker baselines (e.g. MADDPG, QMIX) to the camera ready version. We have so far omitted them since our preliminary experiments suggested that they are unable to learn competitive communication protocols, compared to GME or RL+PR.
>
> > the agent has to take the action that not only serves as to achieve value but also to achieve good communication with the receiver. I wonder what happens if the game includes the redundant action that the agent can use to communicate.
>
> Indeed, the fact that actions must serve simultaneous purposes is a large part of what makes MCGs interesting. If such a property did not hold, the sender would be able to reduce the entropy of the posterior more quickly because the maximum entropy policy would contain additional “free bandwidth”. In our preliminary experiments, we used a version of Pong with redundant actions and did observe this effect.

---

### Official Review · Reviewer_Er8e · 2021-11-08

**Correctness:** 3
**Technical Novelty And Significance:** 2
**Empirical Novelty And Significance:** 1
**Recommendation:** 3
**Confidence:** 3

**Main Review:**

The problem is nice - to keep the same policy and being able to send a message.

**Summary Of The Paper:**

This paper aims to send a message through an MDP. The key is that the policy should still be optimal in the sense of the distribution of actions in each state, while communicating the message.

**Summary Of The Review:**

There are multiple issues in the paper, due to which the reviewer feels that the paper is not ready in its current form.

1. How much message entropy can be communicated? In source coding, we have the entropy of message that is communicated through a certain message length. Similarly, when communicating through a MDP, it is important to know message length communicated for a given finite T length MDP.
2. Since the optimal policy is deterministic, for a given state, there is a unique optimal action, it is unclear how the policy can be the same and still contain the message. This is because deterministic action has no entropy. It seems that the authors are assuming that even if message is sent o(T) of the time, policy will be the same - and thus o(T) times, actions can be chosen - so \tilde{O}(T\log |A| ) length message can be communicated without any change in average policy. Not sure, if the authors are able to get better - and how to quantitatively say that.
3. In general systems, the model of MDP may not be known between sender and receiver. How can these issues be handled?
4. Channel coding and source coding are separate problems, and the authors seem to write the two as same in the text.
5. Two of the special cases are given. How does the results here give same/improved communication results in the two domains. The result of message communications need to be at least the same as in those areas, with results in the general setup.

---

> ### Author Response · Authors · 2021-11-16
> **Thanks for the review!**
>
> We thank the reviewer for their comments.
>
> 1. We are having some trouble parsing this question, but are happy to discuss further if the reviewer wouldn’t mind clarifying.
>
> 2. Indeed, if the MDP policy is deterministic, the sender cannot communicate any information. However, a central point of the submission is that GME’s sender uses MaxEnt RL to compute its MDP policy. For non-zero temperatures, the optimal MaxEnt RL policy is guaranteed to be stochastic. As discussed in section 5.3, the temperature hyper-parameter allows the practitioner to trade-off between the incurred cost from the MDP and (in an informal sense) its communication capacity. For each of the experimental domains, the figures show how performance changes as a function of different temperatures. We hope this addresses your concern, please let us know if we misunderstood the question.
>
> 3. GME does not require the receiver to have access to a model of the MDP. The receiver only requires the sender’s MDP policy and the sender’s minimum entropy coupling scheme.
>
> 4. Thanks for this catch, we did not mean to give this impression. We agree that the paragraph header “Channel Coding” is misnamed. Please let us know if there are other places in the text where this is unclear. We have updated this in the submission.
>
> 5. It’s not obvious to us how one would go about comparing performance across different games. Did the reviewer have something specific in mind?

---

### Official Review · Reviewer_o2vx · 2021-11-08

**Correctness:** 2
**Technical Novelty And Significance:** 3
**Empirical Novelty And Significance:** 2
**Recommendation:** 3
**Confidence:** 4

**Main Review:**

The paper is very clearly written right until the actual method is described. At that point key details vanish and the approach remains half-presented with some ostensible gaps on the receiver side.

In particular, authors take a relatively good care of preparing and presenting the sender's behaviour generation side. From ensuring that the natural policy contains sufficient stochasticity to piggyback a message to some (though fairly naive) theoretical guarantees for the correctness on the sender's side. This reviewer has little argument against that portion. However, the paper completely misses the explanation of \pi_{Z} -- the receiver's decoding policy. The only line about it reads "... the receiver guesses the maximum a posteriori message.".

Unfortunately, posterior calculation, as presented previously in Step 2 belief formulation, depends on the knowledge of the Sender's policy throughout execution. As it is greedily constructed, it seems strange that the receiver would be aware of it to any reasonable extent. This makes the posterior calculation impossible on the receiver's side. One could argue that these are calculated in some form of an equilibrium pattern, but that is not the authors intent or explanation either. So, either there is a secondary communication channel that allows the policy information to be accessible to the receiver -- which contradicts the assumption of communicating by behaviour only -- or there's one half of the approach that authors neglected to present -- which is even more unfortunate.

In addition, authors tend to miss quite a bit of related work. From technological issues (e.g., combinations of RL with information bottlenecks) to ideological "brothers" (e.g., boosting goal recognition in RL solutions and its countermeasures, such as deceptive RL).


**Summary Of The Paper:**

The paper suggests a combination of MaxEnt RL with Minimum Entropy Coupling to construct a communication method via (on the fly) modulation of a stochastic policy execution. The core idea being that communication does not require a separate channel beyond the capability of the receiver to observe the sender's (otherwise goal-driven) behaviour. The paper is supported by several experiments, including a study of external interferences with the established behaviour-based communication channel in the form of action execution uncertainty.


**Summary Of The Review:**

Good core idea and intend, but the paper is incomplete.

---

> ### Author Response · Authors · 2021-11-16
> **Thanks for the review!**
>
> We thank the reviewer for their comments. Our understanding is that the reviewer’s main concern regards how the receiver can compute the posterior when the sender’s policy is defined implicitly by its MDP policy and its MEC algorithm.
>
> The key point here is that the receiver has access to both the trajectory (S^1, A^1, …, S^T, A^T) and the mechanism that the sender uses to construct its policy from its trajectory (MDP policy + MEC algorithm). Therefore, just as the sender (deterministically) constructs its policy online, the receiver can (deterministically) reconstruct the sender’s policy using the same procedure. After doing so, the receiver can compute the posterior. We neglected to mention this procedure in the submission because it is well-established in Dec-POMDP/decentralized stochastic control literature (see references below, for example). That being said, we agree that this explanation deserves more attention than it was given and have added additional detail similar to that above. We have also added additional pseudocode in Algorithm 1 in the appendix. We’d appreciate the reviewer letting us know whether this explanation clears things up or whether additional details are necessary.
>
> We’d also like to emphasize that in our experiments there is no secondary communication channel, beyond the trajectory produced by the sender.
>
> Improving Policies via Search in Cooperative Partially Observable Games (AAAI 2019)\
> Online Planning for Decentralized Stochastic Control with Partial History Sharing (ACC 2019)\
> Solving Common-Payoff Games with Approximate Policy Iteration (AAAI 2021)

---

### Decision · Program_Chairs · 2022-01-20

**Decision:**

Reject

**Comment:**

The paper proposes Markov coding game (MCG), which generalizes both source coding and a large class of referential games. All the reviews are negative. The reviewers think the work is not ready for publication in its current form.